

# Association of dietary inflammatory index with serum asprosin and omentin levels in women with prediabetes

Gizem Taban[1], Nursel Çalık Başaran[2] and Aylin Ayaz[3]

[1] Department of Nutrition and Dietetics, Faculty of Health Sciences, Recep Tayyip Erdoğan University, Rize, Turkey
[2] Department of Internal Diseases, Division of General Internal Medicine, Faculty of Medicine, Hacettepe University, Ankara, Turkey
[3] Department of Nutrition and Dietetics, Faculty of Health Sciences, Hacettepe University, Ankara, Turkey

Corresponding author
Gizem Taban,
gizem.taban@erdogan.edu.tr

## ABSTRACT

**Background and Objectives:** The objective of present study was to assess the relationship between the Dietary Inflammatory Index (DII) and the Glycemic Index (GI), serum TNF-α (tumor necrosis factor alpha), IL-6 (interleukin 6), serum asprosin, and omentin adipokines in prediabetic adult women.

**Methods:** The study included a total of 60 women: 30 women with prediabetes, aged 19–50 years, with a body mass index (BMI) ranging from 25 to 35 kg/m$^2$, and 30 healthy women with similar age and BMI as the control group. Dietary data for calculating DII and GI were obtained from food frequency questionnaires and food consumption records, respectively. Serum levels of asprosin, omentin, IL-6, and TNF-α were analyzed using the enzyme-linked immuno sorbent assay (ELISA) method. Correlation and regression analyses were performed to evaluate the relationships between DII scores, glucose metabolism markers, inflammatory markers, the specified adipokines, and the glycemic index.

**Results:** In the case group, DII scores, GI values, serum asprosin, IL-6, TNF-α, and C-reactive protein (CRP) levels were found to be significantly higher ($p < 0.001$, $p = 0.001$, $p = 0.010$, $p = 0.005$, $p < 0.001$, $p < 0.001$, respectively). No significant difference was found between the case and control groups for serum omentin levels ($p = 0.779$). In the case group, a significant positive correlation was found between DII and insulin, insulin resistance (HOMA-IR) and GI ($r = 0.365$, $p = 0.047$; $r = 0.440$, $p = 0.015$; $r = 0.512$, $p = 0.004$, respectively), but no significant correlation was found with asprosin and omentin ($r = 0.292$, $p = 0.117$; $r = 0.337$, $p = 0.069$, respectively). However, an increase of one unit in serum asprosin levels in the case group was associated with an increase of 0.421 units in the DII score ($F = 6.031$, $p = 0.021$, $\beta = 0.421$, 95% CI [0.008–0.088], adjusted $R^2 = 0.148$), while an increase of one unit in CRP values in the control group was associated with a 0.472 unit increase in the DII score ($F = 8.009$, $p = 0.009$, $\beta = 0.472$, 95% CI [0.343–2.141], adjusted $R^2 = 0.195$).

**Conclusions:** The observed association between increased serum asprosin levels and higher DII scores in prediabetic women may provide preliminary evidence on potential biomarkers for prediabetes, but due to the cross-sectional design of the study, further prospective studies are required to investigate their diagnostic or therapeutic utility. On the other hand, no significant difference was observed between

the groups in terms of serum omentin levels; this may be due to the complexity of the regulation mechanism and requires more detailed investigations.

# INTRODUCTION

Diabetes mellitus (DM) refers to a chronic condition resulting from insufficient synthesis of insulin hormone in the body or inadequate utilization by target tissues (*International Diabetes Federation, 2021*). Type 2 diabetes (T2DM), which is usually triggered by environmental factors such as nutrition, occurs in over 90% of individuals with diabetes (*World Health Organization, 2019*). As reported in the 2021 Atlas of the International Diabetes Federation, over the past 21 years, the number of adults with diabetes globally has increased approximately 3.5 times, reaching 537 million. This number is projected to rise by about 46% by 2045. Türkiye has been shown to have the highest population of people living with diabetes (nine million) in Europe for the year 2021 within this age group (*International Diabetes Federation, 2021*). This situation decreases individuals' quality of life, increases morbidity, mortality, and the economic burden on countries (*Shaw, Sicree & Zimmet, 2010*).

Prediabetes refers to a condition with impaired glucose levels preceding diabetes, which increases the risk of progressing to diabetes (*Echouffo-Tcheugui & Selvin, 2021*). The 2024 National Diabetes Statistics Report of the United States estimates that there are approximately 97.6 million adults with prediabetes (*Centers for Disease Control and Prevention, 2024*). While some individuals with prediabetes may progress T2DM, others may return to normal glucose levels due to the reversible nature of prediabetes (*Tabák et al., 2012*). In the treatment of prediabetes, pharmacological treatment and herbal approaches can be applied in addition to basic lifestyle changes such as nutrition and physical activity (*International Diabetes Federation, 2017*; *Ping et al., 2024*). Obesity and increased insulin resistance in prediabetes trigger an increase in the levels of pro-inflammatory markers, leading to a state of low-grade chronic systemic inflammation (*Hamdy, Porramatikul & Al-Ozairi, 2006*; *Haffner, 2003*). This inflammation contributes to insulin resistance and beta-cell damage (*Rohm et al., 2022*). It has been noted that inflammatory biomarkers, such as serum IL-6, C-reactive protein (CRP), and TNF-α levels, can be influenced by diet (*Shivappa et al., 2017*) and may play a part in the development of chronic inflammation (*Lontchi-Yimagou et al., 2013*). For example, the Mediterranean diet with whole grains, fruits and vegetables, fish, nuts and olive oil was associated with decreased inflammation levels, whereas the Western diet with refined foods, sugar, fried foods, red and processed meat was associated with increased inflammation levels (*Farhangi & Najafi, 2018*; *Mtintsilana et al., 2019*; *Esposito et al., 2004*; *Chrysohoou et al., 2004*; *King, Egan & Geesey, 2003*). Furthermore, studies have highlighted the relationship between diet-induced inflammation and diabetes (*van Woudenbergh et al., 2013*; *Minihane et al., 2015*).

In light of these developments, the Dietary Inflammatory Index (DII) has been created to assess the effect of diet on chronic disease formation based on the relationship between pro- and anti-inflammatory serum markers, macro and micronutrients, and other dietary components (*Shivappa et al., 2014*). Energy, carbohydrates, proteins, total fat, saturated fat, trans fat, cholesterol, iron and vitamin $B_{12}$ have proinflammatory properties, whereas monounsaturated, polyunsaturated fats, omega-3, omega-6, fiber, beta-carotene, folic acid, thiamine, riboflavin, niacin, pyridoxine, retinol, ascorbic acid, tocopherol, vitamin D, magnesium, selenium, zinc, alcohol, caffeine, clove oil, garlic, ginger, onion, turmeric, saffron, green and black tea, flavones, flavonols, flavanones, isoflavones, anthocyanidins, pepper, thyme/marjoram and rosemary have anti-inflammatory properties (*Shivappa et al., 2014*). According to this index, which can be used in different populations, positive or near-positive values reflect the pro-inflammatory effect of the diet, and negative or near-negative values reflect the anti-inflammatory effect of the diet (*Shivappa et al., 2014*). One study found a positive relationship between DII and both the risk and severity of diabetes (*King & Xiang, 2019*). Additionally, a diet that triggers inflammation has been shown to increase the risk of prediabetes in a case-control study (*Vahid et al., 2017*).

Adipokines are defined as molecules derived from adipose tissue, playing roles in various biological mechanisms within the body. However, dysfunction in adipose tissue can adversely affect the adipokines it secretes, potentially leading to various diseases. Therefore, understanding the mechanisms of adipokines may offer new options for diagnosing and treating obesity-related diseases (*Fasshauer & Blüher, 2015*). Asprosin and omentin are two newly identified adipokines, in 2016 and 2006 respectively, both originating from visceral adipose tissue (*Luo et al., 2023*). Asprosin was discovered during a study focused on individuals diagnosed with neonatal progeroid syndrome (NPS). In the fasting state, it is an adipokine secreted by white adipose tissue that triggers the liver to release glucose (*Romere et al., 2016*). Asprosin causes inflammation in pancreatic β cells by stimulating cytokine release. This triggers apoptosis in β cells and leads to decreased insulin secretion (*Lee et al., 2019*). A cross-sectional study found higher plasma asprosin levels in groups with impaired glucose tolerance (IGT) and T2DM compared to a control group with normal glucose tolerance (*Wang et al., 2018*). Omentin (also known as intelectin) is an adipokine with anti-inflammatory and oxidative stress inhibitory properties first identified from human omental fat (*Yang et al., 2006*; *Zhao et al., 2022*). The omentin adipokine has been found to have a positive relationship with insulin sensitivity (*Yang et al., 2006*). Some studies have reported significantly lower serum omentin-1 levels in individuals with T2DM compared to controls (*Zhang et al., 2014*; *Elsaid et al., 2018*), while others have found significantly higher omentin-1 levels in those with T2DM relative to controls (*Madsen et al., 2015*; *Hayashi et al., 2019*).

The Glycemic Index (GI) is defined as the comparison of the glycemic response of a carbohydrate-containing food to that of a glucose solution or white bread containing the same amount of carbohydrate (*Jenkins et al., 1981*). A meta-analysis of studies involving individuals with T2DM has shown that those consuming a diet characterized by a low glycemic index have been more effective in controlling HbA1c and fasting blood glucose (FBG) levels compared to those consuming a high-glycemic index diet or a control group

(*Ojo et al., 2018*). A randomized controlled study found that a diet characterized by a high glycemic index increases serum TNF-α and IL-6 levels (*Kelly et al., 2011*).

In conclusion, different immune response, such as higher levels of Interleukin-1 alpha, have been described in females *vs.* males in the context of inflammatory illnesses, such as autoimmune diseases (*Tripolino et al., 2024*). Prediabetes, a reversible condition, is associated with chronic inflammation (*Tabák et al., 2012*; *Hamdy, Porramatikul & Al-Ozairi, 2006*; *Haffner, 2003*). Therefore, determination of the levels of adipokines such as asprosin and omentin in this period may reflect metabolic and inflammatory change processes. It may also contribute to the identification of new biomarkers for diagnosis (*Diao et al., 2024*; *As'habi et al., 2019*). Although the roles of DII, GI and adipokine levels have been investigated separately in individuals with prediabetes, there are few studies in the literature investigating these variables together in individuals with prediabetes. In the light of the existing literature, a case-control study design was preferred in this study to evaluate the possible associations between DII and GI, serum asprosin, omentin, IL-6 and TNF-α among prediabetic women and age- and body mass index-matched healthy female controls. The chosen study design seems to be suitable for determining exposure differences between individuals with prediabetes and healthy (*Dey, Mukherjee & Chakraborty, 2020*). This study aims to examine of that GI and serum asprosin levels would increase and serum omentin levels would decrease as DII score increased in prediabetic women.

## MATERIALS AND METHODS

### Study sample

This current case-control study was conducted from October 2022 to October 2023 at the policlinics of the Departments of General Internal Medicine and Family Medicine of Hacettepe University Hospitals. Female volunteers aged 19–50 years (the mean age case and control groups were 39.10 ± 7.98 years, 37.57 ± 8.17 years, respectively) with a Body Mass Index (BMI) between 25–35 kg/m$^2$ who were diagnosed with prediabetes according to the American Diabetes Association (ADA) criteria (*American Diabetes Association Professional Practice Committee, 2024*) by a physician at these departments were included in the case group. Volunteers with similar age and BMI who did not have prediabetes were included in the control group. Determination of the study's sample size was computed building the results of prior research, with a Type I error rate (α) of 0.05 and a Type II error rate (β) of 0.20, yielding a statistical power of $1 - \beta = 0.85$. The statistical power analysis was performed using GPower 3.1 software. In all, 60 women were enrolled in the study, with 30 women in the case group and 30 women in the control group. For the case group of prediabetic individuals, participants with glycosylated hemoglobin A1c (HbA1c) levels between 5.7% and 6.4% and/or FBG levels between 100 and 125 mg/dL were included. For the control group, healthy participants with HbA1c <5.7% and FBG <100 mg/dL were accepted (*American Diabetes Association Professional Practice Committee, 2024*). The following exclusion criteria were applied for the study: individuals with Type 1 (T1DM) or T2DM, those using insulin, individuals with acute or chronic complications of diabetes, those using medications affecting glucose tolerance and insulin secretion, individuals

undergoing dietary therapy, pregnant or breastfeeding women, postmenopausal women, individuals with kidney, liver, cardiovascular, thyroid diseases, polycystic ovary syndrome (PCOS), cancer, chronic inflammatory diseases, depression treatment, excessive alcohol consumption (>46 g ethanol/day), chronic smokers, those with acute or chronic infections, individuals taking regular vitamin or mineral supplements in the last 3 months, hormone therapy, long-term anti-inflammatory medications, steroids, antibiotics, and those engaged in heavy physical activity were excluded. The study protocol was granted by the Hacettepe University Non-Interventional Clinical Research Ethics Committee with research number GO 22/1013 on October 18, 2022. Before starting the study, written informed consent was acquired from all participants involved in the study. The Declaration of Helsinki was complied with throughout the research process.

## Anthropometric measurements

Body weight, body composition parameters including lean tissue mass and fat mass were determined by the researcher by means of a TANITA MC180 bioelectrical impedance analysis (BIA) device while participants were barefoot and wearing minimal clothing. The height measurements of the participants were obtained using a stadiometer while they were standing upright, having removed their shoes, with their heads positioned in the Frankfurt plane. BMI (kg/m$^2$) (*World Health Organization, 2000*), waist and hip circumference (cm), and the ratio of the two values to each other (*World Health Organization, 2011*) were measured as specified by the World Health Organization (WHO), and individuals between BMI: 25–35 kg/m$^2$ were involved in the study, according to classification of WHO (*World Health Organization, 2000*).

## Evaluation of dietary intake

Participants' dietary intakes, including dietary components such as energy, macro- and micro-nutrients, were analyzed by transferring the data provided by the quantitative food frequency questionnaire (QFFQ) to the Nutrition Information System 9.0 (BeBiS 9.0) (*Beslenme Bilgi Sistemi, 2021*).

## Calculation of diet-related indices used in the study

The DII is an extensive and current scoring index consisting of 45 food parameters with pro- and anti-inflammatory properties, providing researchers with information about the inflammatory capacity specific to the diets of individuals participating in the study, developed by *Shivappa et al. (2014)*. The daily intake amounts of food parameters for participants were determined using information from the QFFQ, which was then entered into BeBiS 9.0 software. For dietary components not included in BeBiS 9.0 but required for DII calculations, a new database was created with data obtained from the United States Department of Agriculture (USDA) (*Bhagwat, Haytowitz & Holden, 2011*) for flavonoids and the Turkish National Food Composition Database (TürKomp) (*Ulusal Gıda Kompozisyon Veri Tabanı (TürKomp), 2017*) for beta-carotene and integrated into the BeBiS 9.0 programme. In our study, we computed DII using 44 food parameters, with the exception of trans fatty acids, as detailed in earlier studies by *Shivappa et al. (2014)* and
*Toprak et al. (2022).* The GI value of the diet consumed was computed by taking the average of the 2-day food consumption record, one day on weekdays and one day on weekends. The glycemic index value of each food was obtained according to the relevant databases and the glucose values of the foods were calculated with reference (*Foster-Powell, Holt & Brand-Miller, 2002*; *Atkinson, Foster-Powell & Brand-Miller, 2008*). For foods not listed in the tables, the glycemic index value of the most similar food was used (*van Woudenbergh et al., 2011*). High-GI foods are defined as having a GI value ≥70, medium-GI foods as having a GI value 56–69 and low-GI foods as having a GI value ≤55 (*Vega-López, Venn & Slavin, 2018*).

## Serum collection and laboratory measurements

Biochemical parameters such as fasting blood glucose (FBG), glycosylated hemoglobin (HbA1c), fasting insulin, total cholesterol, low-density lipoprotein cholesterol (LDL-C), high-density lipoprotein cholesterol (HDL-C), triglycerides, alanine aminotransferase (ALT), aspartate aminotransferase (AST), and c-reactive protein (CRP) levels obtained common central laboratory procedures were used. Insulin resistance was assessed by obtaining the formula established for the homeostasis model assessment of insulin resistance (HOMA-IR) as shown in a prior study (*Matthews et al., 1985*). For all participants, one tube of blood samples were taken in the early morning following a minimum 10–12 h fast to determine TNF-α, IL-6, asprosin, and omentin levels in serum. The serum was separated from the blood samples by centrifugation at 3,000 rpm for 10 min and stored at −80 °C until analysed. Serum concentrations of asprosin, omentin, IL-6, and TNF-α were measured using enzyme-linked immunosorbent assay (ELISA) kits (Bioassay Technology Laboratory (BT LAB), Shanghai, China), following the manufacturer's protocols. The intra-assay and inter-assay coefficients of variation (CV) were <8% and <10%, respectively. The following catalog numbers were used: asprosin (Cat. No: E4095Hu), omentin (Cat. No: E0155Hu), IL-6 (Cat. No: E0090Hu), and TNF-α (Cat. No: E0082Hu). The assay sensitivity levels, as reported by the manufacturer, were: asprosin: 0.23 ng/mL; omentin: 1.03 ng/L; IL-6: 1.03 ng/L; and TNF-α: 1.52 ng/L.

## Statistical analysis

The research data were analyzed with SPSS (Statistical Package for Social Sciences) Windows version 25.0. The conformity of the data used to the normal distribution was tested by Q–Q Plot drawing. For normally distributed data, differences between two independent groups were examined by independent t-tests, and relationships between continuous variables were analyzed by Pearson correlation analysis. For data not showing normal distribution, differences between two groups were assessed by the Mann–Whitney U test, and relationships between continuous variables were analyzed by Spearman correlation analysis. The similarity of demographic and clinical characteristics between the case and control groups was examined using chi-square analysis. Based on the relationship between DII scores of individuals and serum asprosin, omentin, IL-6, TNF-α and CRP, regression analysis was performed. Initially, these parameters were subjected to regression analysis using the "enter" method. However, due to statistically insignificant results, it was

determined that the "Stepwise" method should be used. All participants were divided into categories according to the tertile cut-off points determined for the entire group in the DII score. Tertiles determined for DII score: T1 < −1.34; T2 (−1.33) − (−1.11); T3 > +1.12. Here, Kruskal-Wallis H test as well as one-way ANOVA test were performed to evaluate the data. A $p$-value of under 0.05 is typically interpreted a threshold for determining statistical significance in data analysis.

## RESULTS

Table 1 displays the basic clinical data for the case and control groups in the study population. The case group's mean age was 39.10 ± 7.98 years, and the control group's mean age was 37.57 ± 8.17 years, no statistically significant difference was observed ($p = 0.465$). No statistically significant difference was found between the case and control groups in height, body weight, BMI, hip circumference, fat mass (%), fat mass (kg), and lean tissue mass ($p = 0.601$, $p = 0.593$, $p = 0.827$, $p = 0.616$, $p = 0.697$, $p = 0.731$, $p = 0.662$, respectively). Waist circumference and waist/hip ratio were found significantly higher in the case group compared to the control group ($p = 0.037$, $p = 0.011$, respectively). In the case group, it was determined that all markers of the glucose metabolism consisting of FBG, HbA1c (%), insulin, HOMA-IR values and LDL-cholesterol, triglyceride, ALT values were significantly higher than those of the control group ($p < 0.001$, $p < 0.001$, $p < 0.001$, $p < 0.001$, $p = 0.001$, $p < 0.001$, $p = 0.003$, respectively), and HDL-cholesterol values were significantly lower than those of the control group ($p = 0.002$). It was determined that serum CRP, asprosin, IL-6, TNF-α, DII and GI values were significantly higher in the case group compared to the control group ($p < 0.001$, $p = 0.010$, $p = 0.005$, $p < 0.001$, $p < 0.001$, $p = 0.001$, respectively), and there was no statistically significant difference between the groups in terms of serum omentin values ($p = 0.779$).

Table 2 displays the correlation between the DII and inflammatory systemic biomarkers, adipokines, GI and glucose metabolism markers in the case and control groups, as well as for the entire study population. In the case group, a significant positive correlation was found between DII and insulin, HOMA-IR, and GI ($r = 0.365$, $p = 0.047$; $r = 0.440$, $p = 0.015$; $r = 0.512$, $p = 0.004$, respectively). For the entire study population, a significant positive correlation was observed between DII and asprosin, IL-6, TNF-α, CRP, HbA1c, FBG, insulin, HOMA-IR, and GI ($r = 0.327$, $p = 0.011$; $r = 0.367$, $p = 0.004$; $r = 0.460$, $p < 0.001$; $r = 0.561$, $p < 0.001$; $r = 0.544$, $p < 0.001$; $r = 0.487$, $p < 0.001$; $r = 0.580$, $p < 0.001$; $r = 0.646$, $p < 0.001$; $r = 0.410$, $p = 0.001$, respectively).

Table 3 shows the relationship between the DII and daily dietary intake of energy, nutrients, and certain flavonoids for both the case and control groups, as well as for the entire study population. A significant negative correlation was found between DII score and energy ($r = −0.540$, $p = 0.002$ and $r = −0.581$, $p = 0.001$) protein ($r = −0.416$, $p = 0.022$ and $r = −0.654$, $p = 0.001$), fat ($r = −0.414$, $p = 0.023$ and $r = −0.409$, $p = 0.025$), carbohydrate ($r = −0.538$, $p = 0.002$ and $r = −0.524$, $p = 0.003$), dietary fiber ($r = −0.780$, $p < 0.001$ and $r = −0.858$, $p < 0.001$), vitamin A ($r = −0.575$, $p = 0.001$ and $r = −0.875$, $p < 0.001$), beta-carotene ($r = −0.702$, $p < 0.001$ and $r = −0.750$, $p < 0.001$), vitamin E ($r = −0.404$, $p = 0.027$ and $r = −0.719$, $p < 0.001$), vitamin B6 ($r = −0.720$, $p < 0.001$ and

**Table 1  Age, anthropometric, biochemical, some adipokine and inflammatory markers, and DII and GI values of the case and control groups.**

| Variables | Case (n = 30) | Control (n = 30) | Test value | p-value |
|---|---|---|---|---|
| Age (years) | 39.10 ± 7.98 | 37.57 ± 8.17 | 0.786[a] | 0.465 |
| Height (m) | 1.61 ± 0.06 | 1.60 ± 0.05 | 0.526[a] | 0.601 |
| Body weight (kg) | 76.59 ± 8.96 | 75.40 ± 8.21 | 0.538[a] | 0.593 |
| BMI (kg/m$^2$) | 29.69 ± 3.23 | 29.51± 3.04 | 0.220 | 0.827 |
| Waist circumference (cm) | 97.23 ± 8.89 | 92.07 ± 9.79 | 2,140[a] | 0.037* |
| Hip circumference (cm) | 111.23 ± 5.93 | 112.12 ± 7.53 | −0.505[a] | 0.616 |
| Waist/hip ratio | 0.88 ± 0.08 | 0.82 ± 0.08 | 2,640[a] | 0.011* |
| Fat mass (%) | 36.16 ± 3.55 | 35.75 ± 4.57 | 0.391[a] | 0.697 |
| Fat mass (kg) | 27.90 ± 5.51 | 27.35 ± 6.84 | 0.345[a] | 0.731 |
| Lean tissue mass (kg) | 48.69 ± 4.38 | 48.24 ± 3.45 | 0.439[a] | 0.662 |
| Fasting blood glucose (mg/dL) | 102.73 ± 10.97 | 85.60 ± 5.76 | 7,573[a] | <0.001** |
| HbA1c (%) | 6.05 ± 0.22 | 5.29 ± 0.26 | 12,212[a] | <0.001** |
| Fasting insulin (uIU/mL) | 13.25 ± 7.20 | 5.70 ± 2.30 | −5,323[b] | <0.001** |
| HOMA-IR | 3.36 ± 1.93 | 1.20 ± 0.46 | −6,209[b] | <0.001** |
| Total cholesterol (mg/dL) | 203.40 ± 42.38 | 185.37 ± 27.49 | −1,575[b] | 0.115 |
| LDL-cholesterol (mg/dL) | 134.27 ± 31.81 | 112.40 ± 19.03 | −3,246[b] | 0.001* |
| HDL-cholesterol (mg/dL) | 50.77 ± 9.50 | 60.13 ± 15.07 | −3,138[b] | 0.002* |
| Triglyceride (mg/dL) | 151.00 ± 64.50 | 77.37 ± 31.92 | 5,604[a] | <0.001** |
| ALT (U/L) | 23.43 ± 9.10 | 16.67 ± 7.70 | 3,109[a] | 0.003* |
| AST (U/L) | 19.63 ± 6.05 | 18.07 ± 4.37 | 1,150[a] | 0.255 |
| CRP (mg/dL) | 0.96 ± 1.18 | 0.42 ± 0.74 | −4,214[b] | <0.001** |
| Asprosin (ng/mL) | 19.95 ± 19.35 | 9.81 ± 4.48 | −2,587[b] | 0.010* |
| Omentin (pg/mL) | 89.93 ± 80.34 | 61.84 ± 26.55 | −0.281[b] | 0.779 |
| IL-6 (pg/mL) | 93.71 ± 113.75 | 37.58 ± 19.97 | −2,794[b] | 0.005* |
| TNF-α (pg/mL) | 151.79 ± 137.49 | 69.15 ± 27.80 | −3,903[b] | <0.001** |
| DII | 1.75 ± 2.22 | −1.39 ± 1.94 | 5,857[a] | <0.001** |
| GI | 56.67 ± 6.05 | 49.38 ± 9.10 | 3,658[a] | 0.001* |

Note:
BMI, body mass index; HBA1C, glycated hemoglobin; HOMA-IR, homeostatic model assessment of insulin resistance; LDL-C, low-density lipoprotein cholesterol; HDL-C, high-density lipoprotein cholesterol; ALT, alanine aminotransferase; AST, aspartate aminotransferase; CRP, c-reactive protein; IL-6, interleukin 6; TNF-a, tumor necrosis factor alpha; DII, dietary inflammatory index; GI, glycemic index, [a]Independent samples t-test, [b]Mann Whitney U test, *p < 0.05, **p < 0.001.

**Table 2  The correlation analysis between the DII and glucose metabolism markers, certain adipokines, inflammatory markers, and GI within both the case and control groups, as well as the total study population.**

| Variables | Case (n = 30) | | Control (n = 30) | | Total (n = 60) | |
|---|---|---|---|---|---|---|
| | r value | p value | r value | p value | r value | p value |
| Fasting blood glucose (mg/dL) | 0.179 | 0.345 | −0.053 | 0.781 | 0.487** | <0.001 |
| HbA1c (%) | 0.031 | 0.869 | 0.099 | 0.604 | 0.544** | <0.001 |
| Insulin (uIU/mL) | 0.365 | 0.047 | 0.032 | 0.867 | 0.580** | <0.001 |

| Variables | Case (n = 30) | | Control (n = 30) | | Total (n = 60) | |
|---|---|---|---|---|---|---|
| | r value | p value | r value | p value | r value | p value |
| HOMA-IR | 0.440 | 0.015 | 0.066 | 0.730 | 0.646** | <0.001 |
| Asprosin (pg/mL) | 0.292 | 0.117 | −0.065 | 0.732 | 0.327* | 0.011 |
| Omentin (pg/mL) | 0.337 | 0.069 | −0.267 | 0.154 | 0.080 | 0.543 |
| IL-6 (pg/mL) | 0.314 | 0.091 | −0.019 | 0.921 | 0.367** | 0.004 |
| TNF-α (pg/mL) | 0.293 | 0.116 | 0.109 | 0.568 | 0.460** | <0.001 |
| CRP (mg/dL) | 0.257 | 0.171 | 0.176 | 0.352 | 0.561** | <0.001 |
| GI | 0.512** | 0.004 | 0.006 | 0.977 | 0.410** | 0.001 |

**Note:**
HbA1c, glycated hemoglobin; HOMA-IR, homeostatic model evaluation of insulin resistance; IL-6, interleukin 6; TNF-a, tumor necrosis factor alpha; CRP, C-reactive protein; GI, glycemic index. Spearman's rank correlation coefficient test, *$p < 0.05$, **$p < 0.01$.

**Table 3 The relationship between individuals' DII scores and daily dietary energy, nutrients and some flavonoids.**

| Variables | Case (n = 30) | | Control (n = 30) | | Total (n = 60) | |
|---|---|---|---|---|---|---|
| | r value | p value | r value | p value | r value | p value |
| **Energy and macronutrients** | | | | | | |
| Energy (kcal/day) | −0.540** | 0.002 | −0.581** | 0.001 | −0.362** | 0.005 |
| Protein (g) | −0.416* | 0.022 | −0.654** | 0.001 | −0.519** | <0.001 |
| Fat (g) | −0.414* | 0.023 | −0.409* | 0.025 | −0.246 | 0.058 |
| Saturated fat (g) | −0.301 | 0.106 | −0.064 | 0.736 | 0.059 | 0.652 |
| MUFA (g) | −0.442* | 0.014 | −0.425* | 0.019 | −0.311* | 0.015 |
| PUFA (g) | −0.463** | 0.010 | −0.580** | 0.001 | −0.378** | 0.003 |
| Omega-3 fatty acid (g) | −0.452* | 0.012 | −0.555** | 0.001 | −0.594** | <0.001 |
| Omega-6 fatty acid (g) | −0.448* | 0.013 | −0.540** | 0.002 | −0.330* | 0.010 |
| Carbohydrate (g) | −0.538** | 0.002 | −0.524** | 0.003 | −0.332** | 0.009 |
| Dietary fiber (g) | −0.780** | <0.001 | −0.858** | <0.001 | −0.852** | <0.001 |
| **Micronutrients (Vitamins)** | | | | | | |
| Vitamin A (μg) | −0.575** | 0.001 | −0.875** | <0.001 | −0.695** | <0.001 |
| Beta-carotene (μg) | −0.702** | <0.001 | −0.750** | <0.001 | −0.809** | <0.001 |
| Vitamin E (mg) | −0.404* | 0.027 | −0.719** | <0.001 | −0.468** | <0.001 |
| Vitamin $B_6$ (mg) | −0.720** | <0.001 | −0.816** | <0.001 | −0.831** | <0.001 |
| Vitamin $B_{12}$ (μg) | −0.306 | 0.100 | −0.223 | 0.236 | −0.160 | 0.223 |
| Total folic acid (μg) | −0.768** | <0.001 | −0.824** | <0.001 | −0.838** | <0.001 |
| Vitamin C (mg) | −0.767** | <0.001 | −0.763** | <0.001 | −0.782** | <0.001 |
| **Micronutrients (Minerals)** | | | | | | |
| Magnesium (mg) | −0.682** | <0.001 | −0.812** | <0.001 | −0.799** | <0.001 |
| Iron (mg) | −0.619** | <0.001 | −0.798** | <0.001 | −0.764** | <0.001 |
| Zinc (mg) | −0.457* | 0.011 | −0.583** | 0.001 | −0.553** | <0.001 |
| Selenium (μg) | −0.343 | 0.063 | −0.142 | 0.455 | −0.180 | 0.168 |

(Continued)

| Variables | Case ($n = 30$) | | Control ($n = 30$) | | Total ($n = 60$) | |
|---|---|---|---|---|---|---|
| | r value | p value | r value | p value | r value | p value |
| **Flavonoids** | | | | | | |
| Flavan-3-ol | −0.323 | 0.082 | −0.323 | 0.082 | −0.373** | 0.003 |
| Flavone | −0.787** | <0.001 | −0.204 | 0.280 | −0.658** | <0.001 |
| Flavonol | −0.687** | <0.001 | −0.730** | <0.001 | −0.781** | <0.001 |
| Flavonone | −0.268 | −0.287 | −0.298 | 0.109 | −0.389** | 0.002 |
| Anthocyanidin | 0.124 | 0.096 | −0.417* | 0.022 | −0.340** | 0.008 |
| Isoflavone | 0.324 | 0.081 | −0.251 | 0.181 | −0.251 | 0.053 |

**Note:**
MUFA, mono-unsaturated fatty acid; PUFA, poly-unsaturated fatty acid. Spearman's rank correlation coefficient test, *$p < 0.05$, **$p < 0.01$.

$r = -0.816$, $p < 0.001$), folic acid ($r = -0.768$, $p < 0.001$ and $r = -0.824$, $p < 0.001$), vitamin C ($r = -0.767$, $p < 0.001$ and $r = -0.763$, $p < 0.001$), magnesium ($r = -0.682$, $p < 0.001$ and $r = -0.812$, $p < 0.001$), iron ($r = -0.619$, $p < 0.001$ and $r = -0.798$, $p < 0.001$), zinc ($r = -0.457$, $p = 0.011$ and $r = -0.583$, $p = 0.001$), monounsaturated fat ($r = -0.442$, $p = 0.014$ and $r = -0.425$, $p = 0.019$), polyunsaturated fat ($r = -0.463$, $p = 0.010$ and $r = -0.580$, $p = 0.001$), omega-3 ($r = -0.452$, $p = 0.012$ and $r = -0.555$, $p = 0.001$), omega-6 ($r = -0.448$, $p = 0.013$ and $r = -0.540$, $p = 0.002$), flavonol ($r = -0.687$, $p < 0.001$ and $r = -0.730$, $p = p < 0.001$) intakes in both the case group and the control group, respectively. In addition, DII score was found to have a significant negative relationship with flavone only in the case group ($r = -0.787$, $p < 0.001$) and anthocyanidin only in the control group ($r = -0.417$, $p = 0.022$). In the entire study population, a significant negative relationship was found between the DII and the intake of energy ($r = -0.362$, $p = 0.005$), protein ($r = -0.519$, $p < 0.001$), carbohydrates ($r = -0.332$, $p = 0.009$), dietary fiber ($r = -0.852$, $p < 0.001$), vitamins A, C, E and B6 ($r = -0.695$, $p < 0.001$; $r = -0.782$, $p < 0.001$; $r = -0.468$, $p < 0.001$; $r = -0.831$, $p < 0.001$) folic acid ($r = -0.838$, $p < 0.001$), magnesium ($r = -0.799$, $p < 0.001$), iron ($r = -0.764$, $p < 0.001$), zinc ($r = -0.553$, $p < 0.001$), monounsaturated fats ($r = -0.311$, $p = 0.015$), polyunsaturated fats ($r = -0.378$, $p = 0.003$), omega-3 ($r = -0.594$, $p < 0.001$), omega-6 ($r = -0.330$, $p = 0.010$), flavan-3-ol ($r = -0.373$, $p = 0.003$), flavone ($r = -0.658$, $p < 0.001$), flavonol ($r = -0.781$, $p < 0.001$), flavanone ($r = -0.389$, $p = 0.002$), anthocyanidin ($r = -0.340$, $p = 0.008$), and beta-carotene ($r = -0.809$, $p < 0.001$).

The results of the regression analysis based on the relationship between individuals' DII scores and biochemical parameters are given in Table 4. In the first model, asprosin in the case group and CRP in the control group were identified as significant predictors in the regression model. It was determined that a one-unit increase in asprosin value increased the DII value by 0.421 units ($F = 6.031$, $p = 0.021$, $\beta = 0.421$, 95% CI [0.008–0.088], adjusted $R^2 = 0.148$), whereas a one-unit increase in CRP value increased the DII value by 0.472 units ($F = 8.009$, $p = 0.009$, $\beta = 0.472$, 95% CI [0.343–2.141], adjusted $R^2 = 0.195$).

**Table 4 Regression analysis results based on the relationship between the individuals' DII scores and biochemical parameters ($n = 60$).**

| Group | Model | Non-standardized | | Standardized | 95% Confidence interval | | $t$ | $p$ value | $F$ value | $p$ value | Adjusted $R^2$ value |
|---|---|---|---|---|---|---|---|---|---|---|---|
| | | Beta | Std. error | Beta | Lower bound | Upper bound | | | | | |
| Case | Fixed | 0.790 | 0.541 | | −0.318 | 1.898 | 1.460 | 0.155 | 6.031 | 0.021* | 0.148 |
| | Asprosin | 0.048 | 0.020 | 0.421 | 0.008 | 0.088 | 2.456 | 0.021 | | | |
| Control | Fixed | −1.914 | 0.367 | | −2.666 | −1.161 | −5.209 | 0.000 | 8.009 | 0.009** | 0.195 |
| | CRP | 1.242 | 0.439 | 0.472 | 0.343 | 2.141 | 2.830 | 0.009 | | | |

Note:

Linear regression. Dependent variable: DII (Dietary Inflammatory Index). CRP: C-reactive protein. $*p < 0.05$, $**p < 0.01$.

**Table 5 Glycemic index, some adipokines and inflammatory markers according to dietary inflammatory index (DII) scores ($n = 60$).**

| Variables | DII tertiles | | | Test value | $p$ value |
|---|---|---|---|---|---|
| | $T_1$ ($n = 20$) (≤−1.34) | $T_2$ ($n = 20$) (−1.33) − (−1.11) | $T_3$ ($n = 20$) (+1.12) | | |
| GI | 49.38 ± 9.53 | 51.85 ± 7.85 | 57.85 ± 5.64 | 6,171[a] | 0.004* (T3 > T1) |
| Asprosin (ng/mL) | 9.12 ± 3.77 | 14.68 ± 12.08 | 20.84 ± 21.23 | 7,319[b] | 0.026* (T3 > T1) |
| Omentin (pg/mL) | 59.37 ± 28.95 | 74.84 ± 54.38 | 93.44 ± 84.52 | 1,259[b] | 0.533 |
| IL-6 (pg/mL) | 34.78 ± 15.26 | 68.49 ± 86.12 | 93.68 ± 115.58 | 8,534[b] | 0.014* (T3 > T1) |
| TNF-α (pg/mL) | 65.07 ± 23.29 | 116.43 ± 109.54 | 149.91 ± 137.98 | 12,054[b] | 0.002* (T3 > T1) |
| CRP (mg/dL) | 0.29 ± 0.17 | 0.64 ± 0.95 | 1.16 ± 1.35 | 18,845[b] | $p < 0.001$* (T3 > T1) |

Note:

DII, dietary inflammatory index; GI, glycemic index; IL-6, interleukin 6; TNF-a, tumor necrosis factor alpha; CRP, C-reactive protein. T1, T2, T3: DII tertiles. T3 > T1: higher in the third group than in the first group. [a]One-way ANOVA test, [b]Kruskal-Wallis H test, $*p < 0.05$.

The distribution of glycemic index, some adipokines and inflammatory markers according to DII score groups is shown in Table 5. According to the DII score groups, GI, asprosin, IL-6, TNF-α, CRP values were found to be significantly higher in the third group than in the first group ($p = 0.004$, $p = 0.026$, $p = 0.014$, $p = 0.002$, $p < 0.001$, respectively). Omentin values were not statistically significant compared to DII groups ($p = 0.533$).

## DISCUSSION

In our study, the case group showed significantly higher levels of DII score, GI value, serum IL-6, asprosin, TNF-α, and CRP, whereas serum omentin levels were higher in the control group, though not significantly. Across the entire study population, a significant positive relationship was found between DII and fasting blood glucose, HbA1c, insulin, HOMA-IR, asprosin, IL-6, TNF-α, CRP levels, and GI. An increase in asprosin in the case group and CRP in the control group was associated with a higher DII score.

In our current study, although the DII value was computed using 44 dietary components, it has been observed that many studies in the literature use fewer parameters, and the number of parameters varies across different studies (*King & Xiang, 2019*; *Vahid et al., 2017*; *Denova-Gutiérrez et al., 2018*). In studies conducted on individuals with T2DM, TNF-α, IL-6 and CRP levels were found to be significantly higher than controls (*Toprak et al., 2022*; *Hu et al., 2004*). Our findings consistent with those of the specified studies, as we found that serum TNF-α, IL-6, and CRP levels were significantly higher in

the case group compared to the control group. Clinical studies and meta-analyses have shown that asprosin levels are elevated in individuals with T2DM and prediabetes compared to healthy controls (*Diao et al., 2024*; *Li et al., 2018*; *Naiemian et al., 2020*; *Wang et al., 2021*; *Ertuna et al., 2023*; *Mahat et al., 2024*; *Ulloque-Badaracco et al., 2024*). It has been suggested that asprosin may play a role in the prediabetic pathway (*Ertuna et al., 2023*). Consistent with these findings, our study also revealed that serum asprosin levels were higher in the case group compared to the control group. However, a study involving T2DM patients reported that plasma omentin-1 levels, before insulin treatment, did not show a significant difference between the case and control groups. In the same study, women were divided into three groups: the control group, a group of obese Type 2 diabetic patients before insulin therapy, and a group after 6 months of insulin therapy. No significant difference in omentin-1 levels was found among these groups (*Komosinska-Vassev et al., 2019*). A meta-analysis evaluating omentin-1 levels in different types of diabetes found that T2DM patients had lower omentin-1 levels compared to controls (*Pan et al., 2019*). However, some studies have reported significantly higher plasma omentin levels in T2DM patients compared to non-diabetic controls (*Madsen et al., 2015*; *Hayashi et al., 2019*). Similarly, in our study, although there was no significant difference between the case and control groups, serum omentin levels were higher in the case group compared to the control group. It has been stated that the lack of difference may be attributed to the complexity of omentin regulation mechanisms (*Komosinska-Vassev et al., 2019*). In a previous study, it was reported that the increase in omentin in the case group could be explained as a compensatory response to inflammation caused by chronic hyperglycemia in diabetes (*Hayashi et al., 2019*). The lack of a significant difference in omentin levels between the case and control groups in our study may be related to this mechanism. In addition, the small sample size may be the cause of the limited statistical relevance of some parameters, including omentin.

Our findings indicate that the DII score is positively associated with insulin and HOMA-IR in the case group, and positively associated with FBG, HbA1c, insulin, and HOMA-IR across all participants. Similar results have been observed in some studies involving individuals with T2DM or prediabetes (*Mtintsilana et al., 2019*; *Shu et al., 2022*; *Zhou et al., 2025*). These studies found a significant positive relationship between DII score and FBG, fasting insulin, HbA1c, and HOMA-IR. However, a study involving healthy adults found no significant relationship between DII score and FBG, fasting insulin, or insulin resistance (*Moslehi et al., 2016*). In a case-control study involving prediabetic individuals, participants with high DII scores had significantly higher fasting plasma glucose (FPG) and HbA1c levels compared to those with low DII scores (*Vahid et al., 2017*). These differing results in the literature may be attributed to variations in study design and the types of patients selected (*Moslehi et al., 2016*).

It was previously determined that the DII score was associated with inflammatory markers IL-6, TNF-α and CRP (*Shivappa et al., 2014*). In addition, a case-control study of obese women with T2DM found a significant positive relationship between DII scores and IL-6, TNF-α, and hs-CRP among all participants (*Toprak et al., 2022*). However, in our study, no significant relationship was found between DII scores and these inflammatory

markers in the case and control groups. This lack of significant association might be attributed to the reduced statistical power due to the small sample size. Nonetheless, when analyzing all participants, a significant positive relationship was identified. Furthermore, in the control group, an increase in serum CRP levels was associated with higher DII scores. Our study also found that IL-6, TNF-α, and CRP levels were significantly higher in the third DII score group (T3) compared to the first group (T1).

In a different study, individuals with high pro-inflammatory DII scores had higher levels of CRP, IL-6, and TNF-α, and lower levels of adiponectin (which has anti-inflammatory properties) compared to those with high anti-inflammatory DII scores (*Phillips et al., 2018*). In our study, serum asprosin levels were significantly higher in the third DII score group compared to the first group, whereas no significant difference was found for serum omentin levels. Additionally, our findings indicate that an increase in serum asprosin levels was associated with a higher DII score in the case group. To our knowledge, there are no studies investigating the relationship between dietary inflammatory load and serum asprosin levels. However, one study has explained that a positive relationship between the DII score and another pro-inflammatory adipokine suggests that the inflammatory properties of the diet might impact adipose tissue inflammation (*Muhammad et al., 2019*). In a study involving healthy obese individuals, it was found that those with high pro-inflammatory DII scores had higher levels of pro-inflammatory chemerin compared to those with low DII scores, although there was no significant difference in omentin levels between groups with different DII scores (*Mirmajidi et al., 2019*). In our study, given that both the case and control groups consisted of mildly overweight and obese individuals, it is thought that the excessive fat accumulation in these individuals might influence the current findings (*Nosrati-Oskouie et al., 2021*). Further research is needed to investigate the relationship between serum omentin levels and the Dietary Inflammatory Index in prediabetic individuals. In the studies conducted, it has been found that individuals with high DII scores have high pro-inflammatory and low anti-inflammatory food parameters, while those with low DII scores have low pro-inflammatory and high anti-inflammatory food parameters (*Denova-Gutiérrez et al., 2018*; *Mirmajidi et al., 2019*; *Saboori et al., 2024*). Generally, our research results are consistent with the findings of these studies.

Studies have indicated that the GI of the diet is related to HbA1c, FBG, and glycemic control in individuals with prediabetes or diabetes (*Zafar et al., 2019*; *Zhu et al., 2021*). A randomized controlled trial found that a low glycemic index diet prevented adverse inflammatory responses (*Gomes, Fabrini & de Alfenas, 2017*). In our study, a significant positive relationship was found between the DII score and GI value in women with prediabetes. There is no literature available comparing both indices in individuals with prediabetes. However, according to the literature, both the DII score and GI value are positively associated with glucose metabolism markers and inflammatory markers, as mentioned above. This may help explain the current relationship between the two indices. Overall, differences in the findings of studies in the literature may be attributed to variations in study populations and designs, participant gender and age, disease type,

diagnostic criteria, the number of dietary components used in DII value calculations, and sample sizes.

In our study, the participants in both the case and control groups were selected to be comparable with regards to age and BMI, and extensive exclusion criteria were implemented to maintain the homogeneity of the study. Since the study included only female participants, an age restriction was implemented to prevent unknown effects of menopausal physiological changes on asprosin. To calculate the DII score, 44 out of 45 food parameters were used, which is more comprehensive than the majority of the literature. Our study is, as far as we know, the first in Türkiye to investigate the relationship between the DII, glycemic index, and asprosin in women with prediabetes. Additionally, given the limited number of clinical studies examining the relationship between DII scores, prediabetes, and the dietary influence on asprosin, our study is expected to make a significant contribution to the literature in this regard.

Due to the cross-sectional design of the study, causality cannot be inferred. For some of the relationships examined, the small size of the sample may have reduced statistical power. The fact that various secondary factors that may affect omentin levels (*e.g.*, age, BMI, physical activity level or regional dietary customs) could not be controlled in our study constitutes a limitation in the interpretation of the results. The results of the DII and GI calculation may also be affected by potential recall bias due to the individual FFQ assessment method. Since the study was conducted only among pre-menopausal Turkish women, the findings may not be generalizable. Although regression analyses are suitable for hypothesis generation, as confounding variables such as physical activity, or socioeconomic influences cannot be adequately accounted for, replication of these findings with larger samples would confirm the exploratory results and provide external validity.

## CONCLUSIONS

The association of the increase in DII scores with the increase in serum asprosin levels in prediabetic women may provide preliminary findings on prediabetes-specific biomarkers, while no significant difference between the groups in terms of serum omentin levels may be due to the complexity of the regulation mechanism. However, since the cross-sectional design of the present study limits the establishment of causality, detailed analyses should be performed in prospective studies with larger sample groups to investigate the diagnostic or therapeutic benefits of these adipokines.

In the following process, it is important to design new studies that can evaluate the practical applicability of the DII score calculation system in prediabetes and other different disease processes and whether diets prepared according to this index benefit individuals. It is important to note that given the irreversible nature of diabetes, its serious complications, decreased quality of life, increased morbidity and mortality risks, and the economic burden on individuals and countries, focusing on prediabetes—an earlier, reversible stage of diabetes—and its management is highly valuable. It is thought that designing studies that will clinically investigate the relationship between the current diets, dietary inflammatory index values and dietary interventions to be applied to these individuals, serum asprosin and omentin adipokines and additionally clearly reveal the biological mechanisms

underlying this condition will enable strategy development and contribute to the literature gap in the current field. In order to confirm the mediating role of adipokines in the metabolic effects of diet, it is recommended that longitudinal or experimental study designs involving anti-inflammatory dietary interventions be favoured in the future.

### Funding
The authors received no funding for this work.

### Competing Interests
The authors declare that they have no competing interests.

### Author Contributions
- Gizem Taban conceived and designed the experiments, performed the experiments, analyzed the data, prepared figures and/or tables, authored or reviewed drafts of the article, and approved the final draft.
- Nursel Çalık Başaran analyzed the data, authored or reviewed drafts of the article, and approved the final draft.
- Aylin Ayaz conceived and designed the experiments, performed the experiments, analyzed the data, prepared figures and/or tables, authored or reviewed drafts of the article, and approved the final draft.

### Human Ethics
The following information was supplied relating to ethical approvals (*i.e.*, approving body and any reference numbers):

TThe study protocol was granted by the Hacettepe University Non-Interventional Clinical Research Ethics Committee with research number GO 22/1013 on October 18, 2022.

### Data Availability
Raw data is available in the Supplemental File.

### Supplemental Information
Supplemental information for this article can be found online at http://dx.doi.org/10.7717/peerj.19957#supplemental-information.

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

# PeerJ

**Madsen SM, Thorup AC, Bjerre M, Jeppesen PB. 2015.** Does 8 weeks of strenuous bicycle exercise improve diabetes-related inflammatory cytokines and free fatty acids in type 2 diabetes patients and individuals at high-risk of metabolic syndrome? *Archives of Physiology and Biochemistry* **121(4)**:129–138 DOI 10.3109/13813455.2015.1082600.

**Mahat RK, Jantikar AM, Rathore V, Panda S. 2024.** Circulating asprosin levels in type 2 diabetes mellitus: A systematic review and meta-analysis. *Clinical Epidemiology and Global Health* **25**:101502 DOI 10.1016/j.cegh.2023.101502.

**Matthews DR, Hosker JR, Rudenski AS, Naylor BA, Treacher DF, Turner RC. 1985.** Homeostasis model assessment: insulin resistance and fl-cell function from fasting plasma glucose and insulin concentrations in man. *Diabetologia* **28(7)**:412–419 DOI 10.1007/BF00280883.

**Minihane AM, Vinoy S, Russell WR, Baka A, Roche HM, Tuohy KM, Teeling JL, Blaak EE, Fenech M, Vauzour D, McArdle HJ, Kremer BHA, Sterkman L, Vafeiadou K, Benedetti MM, Williams CM, Calder PC. 2015.** Low-grade inflammation, diet composition and health: current research evidence and its translation. *The British Journal of Nutrition* **114(7)**:999–1012 DOI 10.1017/s0007114515002093.

**Mirmajidi S, Izadi A, Saghafi-Asl M, Vahid F, Karamzad N, Amiri P, Shivappa N, Hébert JR. 2019.** Inflammatory potential of diet: association with chemerin, omentin, lipopolysaccharide-binding protein, and insulin resistance in the apparently healthy obese. *Journal of the American College of Nutrition* **38(4)**:302–310 DOI 10.1080/07315724.2018.1504348.

**Moslehi N, Ehsani B, Mirmiran P, Shivappa N, Tohidi M, Hébert J, Azizi F. 2016.** Inflammatory properties of diet and glucose-insulin homeostasis in a cohort of Iranian adults. *Nutrients* **8(11)**:735 DOI 10.3390/nu8110735.

**Mtintsilana A, Micklesfield LK, Chorell E, Olsson T, Shivappa N, Hebert JR, Kengne AP, Goedecke JH. 2019.** Adiposity mediates the association between the dietary inflammatory index and markers of type 2 diabetes risk in middle-aged black South African women. *Nutrients* **11(6)**:1246 DOI 10.3390/nu11061246.

**Muhammad HFL, van Baak MA, Mariman EC, Sulistyoningrum DC, Huriyati E, Lee YY, Wan Muda WAM. 2019.** Dietary inflammatory index score and its association with body weight, blood pressure, lipid profile, and leptin in Indonesian adults. *Nutrients* **11(1)**:148 DOI 10.3390/nu11010148.

**Naiemian S, Naeemipour M, Zarei M, Lari Najafi M, Gohari A, Behroozikhah MR, Heydari H, Miri M. 2020.** Serum concentration of asprosin in new-onset type 2 diabetes. *Diabetology & Metabolic Syndrome* **12**:65 DOI 10.1186/s13098-020-00564-w.

**Nosrati-Oskouie M, Asghari G, Yuzbashian E, Aghili-Moghaddam NS, Zarkesh M, Safarian M, Mirmiran P. 2021.** Does dietary intake impact omentin gene expression and plasma concentration? A systematic review. *Lifestyle Genomics* **14(2)**:49–61 DOI 10.1159/000513885.

**Ojo O, Ojo OO, Adebowale F, Wang XH. 2018.** The effect of dietary glycaemic index on glycaemia in patients with type 2 diabetes: a systematic review and meta-analysis of randomized controlled trials. *Nutrients* **10(3)**:373 DOI 10.3390/nu10030373.

**Pan X, Kaminga AC, Wen SW, Acheampong K, Liu A. 2019.** Omentin-1 in diabetes mellitus: a systematic review and meta-analysis. *PLOS ONE* **14(12)**:e0226292 DOI 10.1371/journal.pone.0226292.

**Phillips CM, Shivappa N, Hébert JR, Perry IJ. 2018.** Dietary inflammatory index and biomarkers of lipoprotein metabolism, inflammation and glucose homeostasis in adults. *Nutrients* **10(8)**:1033 DOI 10.3390/nu10081033.

**Ping WX, Hu S, Su JQ, Ouyang SY. 2024.** Metabolic disorders in prediabetes: from mechanisms to therapeutic management. *World Journal of Diabetes* **15(3)**:361–377 DOI 10.4239/wjd.v15.i3.361.

**Rohm TV, Meier DT, Olefsky JM, Donath MY. 2022.** Inflammation in obesity, diabetes, and related disorders. *Immunity* **55(1)**:31–55 DOI 10.1016/j.immuni.2021.12.013.

**Romere C, Duerrschmid C, Bournat J, Constable P, Jain M, Xia F, Saha PK, Del Solar M, Zhu B, York B, Sarkar P, Rendon DA, Gaber MW, LeMaire SA, Coselli JS, Milewicz DM, Sutton VR, Butte NF, Moore DD, Chopra AR. 2016.** Asprosin, a fasting-induced glucogenic protein hormone. *Cell* **165(3)**:566–579 DOI 10.1016/j.cell.2016.02.063.

**Saboori S, Mousavi N, Vahid F, Hebert JR, Asbaghi O, Choobkar S, Birjandi M, Bahram FT, Rad EY. 2024.** The Association of Dietary Inflammatory Index with the risk of type 2 diabetes: a case-control study. *Journal of Nutrition and Food Security* **9(3)**:413–422 DOI 10.18502/jnfs.v9i3.16151.

**Shaw JE, Sicree RA, Zimmet PZ. 2010.** Global estimates of the prevalence of diabetes for 2010 and 2030. *Diabetes Research and Clinical Practice* **87(1)**:4–14 DOI 10.1016/j.diabres.2009.10.007.

**Shivappa N, Hebert JR, Marcos A, Diaz LE, Gomez S, Nova E, Michels N, Arouca A, González-Gil E, Frederic G, González-Gross M, Castillo MJ, Manios Y, Kersting M, Gunter MJ, De Henauw S, Antonios K, Widhalm K, Molnar D, Moreno L, Huybrechts I. 2017.** Association between dietary inflammatory index and inflammatory markers in the HELENA study. *Molecular Nutrition & Food Research* **61(6)**:1–10 DOI 10.1002/mnfr.201600707.

**Shivappa N, Steck SE, Hurley TG, Hussey JR, Hébert JR. 2014.** Designing and developing a literature-derived, population-based dietary inflammatory index. *Public Health Nutrition* **17(8)**:1689–1696 DOI 10.1017/s1368980013002115.

**Shu Y, Wu X, Wang J, Ma X, Li H, Xiang Y. 2022.** Associations of dietary inflammatory index with prediabetes and insulin resistance. *Frontiers in Endocrinology* **13**:820932 DOI 10.3389/fendo.2022.820932.

**Tabák AG, Herder C, Rathmann W, Brunner EJ, Kivimäki M. 2012.** Prediabetes: a high-risk state for diabetes development. *The Lancet* **379(9833)**:2279–2290 DOI 10.1016/s0140-6736(12)60283-9.

**Toprak K, Görpelioğlu S, Özsoy A, Özdemir Ş, Ayaz A. 2022.** Does fetuin-A mediate the association between pro-inflammatory diet and type-2 diabetes mellitus risk? *Nutricion Hospitalaria* **39(2)**:383–392 DOI 10.20960/nh.03848.

**Tripolino O, Mirabelli M, Misiti R, Torchia A, Casella D, Dragone F, Chiefari E, Greco M, Brunetti A, Foti DP. 2024.** Circulating autoantibodies in adults with Hashimoto's thyroiditis: new insights from a single-center, cross-sectional study. *Diagnostics* **14(21)**:2450 DOI 10.3390/diagnostics14212450.

**Ulloque-Badaracco JR, Al-kassab-Córdova A, Hernandez-Bustamante EA, Alarcon-Braga EA, Robles-Valcarcel P, Huayta-Cortez MA, Cabrera Guzmán JC, Seminario-Amez RA, Benites-Zapata VA. 2024.** Asprosin levels in patients with type 2 diabetes mellitus, metabolic syndrome and obesity: a systematic review and meta-analysis. *Diabetes and Metabolic Syndrome: Clinical Research and Reviews* **18(7)**:103095 DOI 10.1016/j.dsx.2024.103095.

**Ulusal Gıda Kompozisyon Veri Tabanı (TürKomp). 2017.** T.C. Tarım ve Orman Bakanlığı. *Available at* https://turkomp.tarimorman.gov.tr/component_result-beta-karoten-45 (accessed 25 February 2024).

**Vahid F, Shivappa N, Karamati M, Naeini AJ, Hebert JR, Davoodi SH. 2017.** Association between Dietary Inflammatory Index (DII) and risk of prediabetes: a case-control study. *Applied Physiology, Nutrition, and Metabolism* **42(4)**:399–404 DOI 10.1139/apnm-2016-0395.

**van Woudenbergh GJ, Kuijsten A, Sijbrands EJG, Hofman A, Witteman JCM, Feskens EJM. 2011.** Glycemic index and glycemic load and their association with C-reactive protein and incident type 2 diabetes. *Journal of Nutrition and Metabolism* **2011**:623076 DOI 10.1155/2011/623076.

**van Woudenbergh GJ, Theofylaktopoulou D, Kuijsten A, Ferreira I, Van Greevenbroek MM, Van Der Kallen CJ, Schalkwijk CG, Stehouwer CD, Ocké MC, Nijpels G, Dekker JM, Blaak EE, Feskens EJM. 2013.** Adapted dietary inflammatory index and its association with a summary score for low-grade inflammation and markers of glucose metabolism: the Cohort study on Diabetes and Atherosclerosis Maastricht (CODAM) and the Hoorn study. *The American Journal of Clinical Nutrition* **98(6)**:1533–1542 DOI 10.3945/ajcn.112.056333.

**Vega-López S, Venn BJ, Slavin JL. 2018.** Relevance of the glycemic index and glycemic load for body weight, diabetes, and cardiovascular disease. *Nutrients* **10(10)**:1361 DOI 10.3390/nu10101361.

**Wang R, Lin P, Sun H, Hu W. 2021.** Increased serum asprosin is correlated with diabetic nephropathy. *Diabetology & Metabolic Syndrome* **13**:51 DOI 10.1186/s13098-021-00668-x.

**Wang Y, Qu H, Xiong X, Qiu Y, Liao Y, Chen Y, Zheng Y, Zheng H. 2018.** Plasma asprosin concentrations are increased in individuals with glucose dysregulation and correlated with insulin resistance and first-phase insulin secretion. *Mediators of Inflammation* **2018**:9471583 DOI 10.1155/2018/9471583.

**World Health Organization. 2000.** *Obesity: preventing and managing the global epidemic: report of a WHO consultation*. Geneva: World Health Organization. *Available at* https://iris.who.int/handle/10665/42330 (accessed 10 September 2024).

**World Health Organization. 2011.** *Waist circumference and waist-hip ratio: report of a WHO expert consultation*. Geneva: World Health Organization, 8–11 December 2008. *Available at* https://iris.who.int/handle/10665/44583.

**World Health Organization. 2019.** *Classification of diabetes mellitus*. Geneva: World Health Organization. Licence: CC BY-NC-SA 3.0 IGO. *Available at* https://iris.who.int/bitstream/handle/10665/325182/9789241515702-eng.pdf.

**Yang R-Z, Lee M-J, Hu H, Pray J, Wu H-B, Hansen BC, Shuldiner AR, Fried SK, McLenithan JC, Gong D-W. 2006.** Identification of omentin as a novel depot-specific adipokine in human adipose tissue: possible role in modulating insulin action. *American Journal of Physiology-Endocrinology and Metabolism* **290(6)**:E1253–E1261 DOI 10.1152/ajpendo.00572.2004.

**Zafar MI, Mills KE, Zheng J, Regmi A, Hu SQ, Gou L, Chen L-L. 2019.** Low-glycemic index diets as an intervention for diabetes: a systematic review and meta-analysis. *The American Journal of Clinical Nutrition* **110(4)**:891–902 DOI 10.1093/ajcn/nqz149.

**Zhang Q, Zhu L, Zheng M, Fan C, Li Y, Zhang D, He Y, Yang H. 2014.** Changes of serum omentin-1 levels in normal subjects, type 2 diabetes and type 2 diabetes with overweight and obesity in Chinese adults. *Annales d'Endocrinologie* **75(3)**:171–175 DOI 10.1016/j.ando.2014.04.013.

**Zhao A, Xiao H, Zhu Y, Liu S, Zhang S, Yang Z, Du L, Li X, Niu X, Wang C, Yang Y, Tian Y. 2022.** Omentin-1: a newly discovered warrior against metabolic related diseases. *Expert Opinion on Therapeutic Targets* **26(3)**:275–289 DOI 10.1080/14728222.2022.2037556.

**Zhou Q, Gao Y, Guo Y, Zhu G. 2025.** Relationship between dietary inflammatory index and blood glucose changes in patients with pre-diabetes mellitus. *Alternative Therapies in Health and Medicine* **31(1)**:276–282.

**Zhu R, Larsen TM, Fogelholm M, Poppitt SD, Vestentoft PS, Silvestre MP, Jalo E, Navas-Carretero S, Huttunen-Lenz M, Taylor MA, Stratton G, Swindell N, Drummen M, Adam TC, Ritz C, Sundvall J, Valsta LM, Muirhead R, Brodie S, Handjieva-Darlenska T, Handjiev S, Martinez JA, Macdonald IA, Westerterp-Plantenga MS, Brand-Miller J, Raben A. 2021.** Dose-dependent associations of dietary glycemic index, glycemic load, and fiber with 3-year weight loss maintenance and glycemic status in a high-risk population: a secondary analysis of the diabetes prevention study PREVIEW. *Diabetes Care* **44(7)**:1672–1681 DOI 10.2337/dc20-3092.