# Peer review of "Association of dietary inflammatory index with serum asprosin and omentin levels in women with prediabetes"

_PeerJ, doi:10.7717/peerj.19957_

## Round 0.1 · original submission · Major Revisions

I invite the authors to carefully revise their manuscript according to the reviewers' concerns.

I would like to point out a particular attention to the following:
the study involves a small number of participants, therefore I would indicate it as a pilot study. The small sample size may be the cause of the limited statistical relevance of some parameters, including omentin (this can be added to the discussion).

Please indicate the analytical performance of the ELISA tests (intra- and inter-assay CVs) in the methods, and for the routine assays, change "from patients files" indicating that common central laboratory procedures were used.

As already indicated by a reviewer, please claim that the DII comprises analytes that are not specific for dietary inflammation, but generic inflammatory markers that have been clustered by previous studies under this term (if necessary, add references). At present, no specific dietary inflammation biomarkers exist.

Finally, I appreciated that in your study, even if the sample size is small, patients and controls look well-matched. Limiting to the female sex (at premenopausal age) can be seen as a good choice in the experimental plan, as different immune response, such as higher levels of Interleukin-1 alpha, have been described in females vs. males in the context of inflammatory illnesses, such as autoimmune diseases (Tripolino O, et al. Diagnostics, 2024).

Reviewer 1 ·

Basic reporting

The article is quite good and delivers the result, although several mistakes need to be revised.

Experimental design

A well-defined, sufficient elaboration

Validity of the findings

Several issues were found in statistical writing and need to be clarified.

Annotated reviews are not available for download in order to protect the identity of reviewers who chose to remain anonymous.

Reviewer 2 ·

Basic reporting

1. While the study focuses on females, the introduction lacks a clear justification for this choice. Given that globally, the prevalence of prediabetes is often higher in males, the authors should provide a rationale for specifically selecting females as the population of interest.
2. Lines 54–55: Please expand the discussion on treatment options for prediabetes beyond exercise. Not all individuals can engage in regular physical activity due to physical or environmental limitations. Include pharmacological and herbal treatments as alternatives.
3. Lines 59–60: The phrase “dietary inflammatory biomarkers” may be misleading. Consider revising it to “inflammatory biomarkers, such as serum IL-6, CRP, and TNF-alpha”, as these markers are not inherently “dietary,” but rather respond to dietary influences.
4. Lines 56–57: The section is insufficiently supported by references. Please include additional relevant literature.

Experimental design

1. While the definition of GI is provided in the introduction, it would be helpful for readers if the authors also explain in the Methods section how GI values are interpreted in this study. For example, briefly include the classification thresholds (e.g., low ≤ 55, medium 56–69, high ≥ 70).
2. To improve reproducibility and transparency, please include the catalogue numbers and manufacturers of any assay kits used for biomarker measurements.
3. Line 105: Ensure that the American Diabetes Association (ADA) guideline cited is properly referenced. Include the full reference or DOI to enhance traceability.

Validity of the findings

No comment

Additional comments

1. The current discussion contains repetitive statistical data, which duplicates what is already presented in the results section. Consider reducing the numerical repetition and instead summarize key trends and findings narratively.
2. Lines 245–248: The first paragraph of the discussion may be removed. Instead, the discussion should begin by summarizing the study's main findings, followed by comparison with existing literature, and then a biological explanation (mechanism) for the observed results.
3. Revise the discussion to place greater emphasis on the biological mechanisms behind your findings. Avoid simply restating the statistical outcomes, instead, develop a clear narrative that contextualizes your results within the broader scientific literature.

Reviewer 3 ·

Basic reporting

Title & Abstract
Title:
The manuscript presents two different titles, “Association of dietary inflammatory index with serum
asprosin and omentin levels in women with prediabetes” and “Is dietary inflammatory index associated with serum asprosin and omentin levels in women with prediabetes?”, which is confusing. The authors need to consider one title.
However, the title can be improved further by highlighting the associations with glycemic index (GI) and other inflammatory markers (CRP, IL-6, TNF-α).
Associations of Dietary Inflammatory and Glycemic Indices with Serum Asprosin, Omentin, and Inflammatory Biomarkers in Women with Prediabetes
Abstract:
Please provide a structured abstract. Although the abstract effectively summarizes the methods and findings of the study, it marginalizes the discussion of omentin. The non-significant result for omentin is important and should perhaps be more explicitly contextualized. The authors are requested to note that the implications related to prediabetes diagnosis and treatment could be worded more cautiously to avoid overstatement, as causality cannot be confirmed in a case-control study.
If word limit permits, consider adding a sentence on limitations or the interpretability of cross-sectional associations.
The authors are requested to be consistent in reporting key analytical results (e.g., always include effect sizes or correlation coefficients, not just p-values).

Introduction
The current version of the manuscript justifies the focus of the study on prediabetes, a critical metabolic state, and highlights the significance of pro-inflammatory diets in fueling systemic inflammation and metabolic dysregulation. It also demonstrated the role of dietary inflammatory index (DII) and glycemic index (GI) in modulating inflammation and insulin resistance, offering appropriate links to asprosin and omentin as adipokines. The introduction effectively demonstrates the emerging role of asprosin in glucose metabolism and inflammation, highlighting its relevance to prediabetes.
The authors have also described Omentin as an anti-inflammatory adipokine, positioning it as a counterpoint to asprosin, framing the novelty of examining their interplay in the context of inflammatory diets. References to prior studies are adequately provided, which underline gaps in understanding the DII-biochemical marker relationship, establishing the aim of the study.
The authors are requested to briefly explain the specific components, such as nutrients negatively or positively associated with inflammation, to educate the audience on their calculation and practical relevance.
The authors are also requested to incorporate an explanation of human dietary patterns, such as fast foods, refined carbohydrates, linked to high DII/GI scores in prediabetic populations that would contextualize the results in real-world implications. The authors need to emphasize why asprosin and omentin require further study in prediabetes. As these adipokines are linked to glycemic control, what specific knowledge gap or clinical importance does this study address? It is recommended to tailor the role of these adipokines to chronic low-grade inflammation found in prediabetic subjects might help consolidate the narrative
The authors are requested to contextualize the reason for selecting the case-control study design in the introduction and why this approach is appropriate for revealing the associations, which will help the readers relate to the methodology.
It is important to mention how covariate factors, such as age, BMI, physical activity, or regional dietary customs, could influence the biomarkers. This would help frame the broader applicability of findings.
While the authors have outlined the study aim, specific hypotheses (e.g., higher DII and GI scores correlate positively with asprosin levels and negatively with omentin) could guide the reader better, and clarifying expectations of whether DII affects insulin resistance via adipokines and inflammatory markers would strengthen the conceptual framework.

Figures & Tables
The tables look fine. There are no figures.

Experimental design

Materials and Methods
The methods section of the manuscript provides a detailed description of study procedures, which is well-structured and clear. The authors have mentioned the rationale for selecting prediabetic women aged 19–50 years with body mass indices (BMI) between 25–35 kg/m2, which reflects a focused approach.
The extensive inclusion/exclusion criteria set by the authors have ensured homogeneity of the study population. For example, participants with chronic diseases, specific medication use, or dietary interventions were excluded, which reduced the chances of a confounding effect.
The authors used validated tools, such as food frequency questionnaires (FFQ), enriched with databases like BeBiS 9.0, to calculate dietary scores, which substantiated their objectives. Use of standardized experimental techniques, such as bioelectrical impedance analysis for body composition, added to the methodological rigor.
The authors also comprehensively studied the combination of biochemical, inflammatory, and dietary indices (e.g., asprosin, omentin, CRP, DII, GI) that effectively operationalized the objectives of the present study.
The detailed description of the subject recruitment, data collection, laboratory methods, and statistical analysis in the methodology of the study enhanced its replicability.
However, there are certain concerns regarding the methodology that need to be addressed, like:
1. Mention specific reference ranges or assay sensitivity levels for measures like asprosin could clarify the applicability between lines 158 to 170, page no.
2. Mention of how dietary data translation between databases (BeBiS, USDA, TürKomp) was standardized to improve transparency between lines 138-157.
The study was conducted from October 2022 to October 2023, which is sufficient for observing cross-sectional relationships in prediabetic biomarkers and dietary patterns. However, it is imperative to highlight that single-timepoint evaluation in cross-sectional study designs inherently limits conclusions about causality or longitudinal effects of diet in the limitations within the discussion section.
The small sample size of 60 individuals could underpower the subgroup analysis, limiting the conclusions. Therefore, the authors need to address this as a limitation of the study. Furthermore, stepwise regression could yield significant associations, but it might increase susceptibility to overfitting. These analyses should be cross-validated or supplemented with justifications for variable inclusion/exclusion.
The authors are requested to mention that self-reported FFQs are prone to recall bias, influencing DII and GI calculations. The adjustments for confounders (e.g., physical activity, socioeconomic status) are under-explained in the text for the regression models, which is necessary for inferring conclusions.
It is recommended to include a sensitivity analysis (e.g., bootstrapping) or an adjustment for multiple testing to strengthen the study conclusions.
The authors need to mention the gender-constrained findings, such as menopausal effects, and generalizability to male or non-Turkish populations would contextualize the results.

Validity of the findings

Results
This is an interesting study to read, where the study addressed an emerging area of interest and examined the relationship between dietary inflammatory indices (DII, GI) and serum adipokines (asprosin, omentin) along with inflammatory markers (e.g., CRP, IL-6, TNF-α) in a prediabetic population. The novelty of the study lies in the findings that asprosin and omentin are studied in the context of diet-induced inflammation in prediabetes, which is an under-researched area, making the study a worthwhile addition to the scientific literature. The authors have integrated dietary indices, novel biomarkers, and prediabetic metabolic states, bridging nutritional and endocrinologic research. Focusing on women with prediabetes as a target group represents an important effort to identify early-stage biomolecular changes and gender specific trends.
However, the small sample size and lack of longitudinal data constrain broader generalizations and also under power subgroup analyses.
The overall data seems credible and plausible in reducing the knowledge gap in this research niche. However, the non-significant association of omentin with DII contrasts with prior findings of decreased omentin in inflammatory conditions (Ref: Dooxa Nongrum A, Guru SR, K J N, Aghanashini S. Analysing adipokine Omentin-1 in periodontal disease and type-2 diabetes mellitus: An interventional comparative study. Journal of Oral Biology and Craniofacial Research. 2022;12(2):273-278. doi:10.1016/j.jobcr.2022.03.010), raising questions about statistical power or the sensitivity of the assays in detecting subtle relationships.
The self-reported dietary data are inherently prone to bias, potentially attenuating observed relationships.
The regression analyses, while appropriate for hypothesis generation, did not adequately account for confounders like physical activity, hormonal factors, or socioeconomic influences. The authors are requested to include the variables and perform the analysis, or the authors can address this by mentioning that replicating findings with larger, more diverse cohorts would confirm the exploratory insights while supporting external validity, in the discussion section. In future research, employing longitudinal or interventional designs, such as using anti-inflammatory diets to substantiate the role of adipokines as dietary mediators.
Secondary factors influencing omentin levels need to be considered in the analysis, if possible, else they can be discussed in the discussion section between lines 262-275 to contextualize the findings.
The authors are requested to implement more robust adjustment strategies or sensitivity analyses to mitigate biases or reveal masked associations. In the text, the p-values should be accompanied by the summary stats, like the effect sizes, confidence intervals.
The low R2 values of the regression model indicate that the model is not fit with limited explanatory power. It is recommended to incorporate additional predictors for multivariate analyses. The omission of trans-fats from the DII indicates incompleteness of the dietary assessments.
Furthermore, the lack of observed differences in omentin across groups could be due to underreporting or variability in the experimental assays or sample representation. These points to the need for future research on large and diverse populations to capture the complex interplay of adipokines in chronic glycemic states.
The authors need to extend the CRP and IL-6 analysis into tertile-based comparisons. The observation of high variance in TNF-α indicates potential instability in the data and points towards the secondary indicators of inflammation. Finally, the quantification of micronutrient data is necessary when the BeBis database is used to interpolate data on nutrients, such as flavonoids.

Discussion
The discussion is well written. It is contextualized and correlated with the findings of the study with proper citation. The lack of association of omentin is addressed in the discussion, but the findings need to be highlighted in the results.
One observation is that the discussion is sub headed. It is recommended to remove the subheadings.

Conclusion
The authors have implied a causal relationship between dietary inflammatory profiles and metabolic dysfunction, which in a cross-sectional design becomes an overestimation of the evidence. The authors have touched upon the early metabolic changes as predictive markers of diabetes progression. However, the authors should emphasize the DII-induced alterations in adipokine levels, which specifically enhance the risk of diabetes, to strengthen the translational relevance of the study conducted. The authors should address the feasibility of translating the findings on the DII-induced adipokine levels into clinical practice, where asprosin could be used as a biomarker.

---

## Round 0.2 · accepted · Accept

The authors have satisfactorily addressed all the issues raised by the reviewers.
In particular, reviewers 2 and 3 have re-reviewed and recommended acceptance, while the Editor has checked answers to the notes from reviewer 1 and her own comments for the positive, final decision.

Reviewer 2 ·

Basic reporting

Comments adressed

Experimental design

Comments addressed

Validity of the findings

Comments adressed

Reviewer 3 ·

Basic reporting

Title & Abstract
The authors have adequately addressed the comments in the title and abstract sections. The authors have revised the title. The authors have adequately added the summary metrics in the results section, structured the abstract, and added the conclusion. The title and the abstract are well revised by the authors, and it looks good.

Introduction
The authors have addressed the suggestions given in the Introduction. They have adequately established the link between dietary patterns and DII/GI scores in the prediabetic population. The authors have sufficiently contextualized the nutrients linked with inflammation, the background evidence and reason for selecting the case–control study design, and elaborated the knowledge gaps with literature evidence. The introduction has improved as the authors have satisfactorily revised the text.

Figures & Tables
The tables are adequately presented.

Experimental design

Material and Methods
The authors have adequately addressed all the comments with appropriate changes made in the text as suggested. The clarifications provided by the authors on certain tests are justified and acceptable. The rationale is correct for not performing the sensitivity analysis, and the covariate adjustments are justified properly.

Validity of the findings

Results
The authors have adequately addressed the comments in the results and clearly justified the rationale behind each of the experiments and findings. The authors have also removed the repetitions and revised the statistical interpretations.

Discussion
The authors have adequately addressed all the comments on the discussion, contextualizing the findings with current literature evidence, and have removed the repetitions of the results.

Conclusion
The authors have addressed the comments satisfactorily.